# Effects of the Pharmacological Modulation of NRF2 in Cancer Progression

**DOI:** 10.3390/medicina61122224

**Published:** 2025-12-16

**Authors:** Santiago Gelerstein-Claro, Gabriel Méndez-Valdés, Ramón Rodrigo

**Affiliations:** Interdisciplinary Center for Pharmacology and Immunology, Institute of Biomedical Sciences, Faculty of Medicine, University of Chile, Santiago 8380000, Chile; santiago.gelerstein@ug.uchile.cl (S.G.-C.); gabrielmendez@ug.uchile.cl (G.M.-V.)

**Keywords:** NRF2, KEAP1, tumor invasion, metastasis, pharmacologic modulation, metabolic reprogramming

## Abstract

Nuclear factor erythroid 2-related factor 2 (NRF2) orchestrates redox balance, metabolism, and cellular stress responses, acting as both a tumor suppressor and promoter depending on the disease stage. In advanced cancers, persistent NRF2 activation—through KEAP1/NFE2L2 mutations or oxidative adaptation—drives epithelial-to-mesenchymal transition, metabolic reprogramming, and immune evasion, promoting tumor invasion (T) and metastasis (M). Recent pharmacologic efforts seek to exploit this duality. NRF2 inhibitors such as brusatol, halofuginone, and ML385 suppress NRF2 transcriptional activity or disrupt DNA binding, reducing motility, invasion, and metastatic dissemination in preclinical models. In contrast, NRF2 activators, such as bardoxolone methyl (CDDO-Me), sulforaphane, and dimethyl fumarate, exhibit chemopreventive effects by enhancing detoxification and mitigating oxidative DNA damage during early tumorigenesis. Furthermore, metabolic interventions, such as glutaminase or G6PD inhibitors, target NRF2-driven anabolic and antioxidant pathways essential for metastatic fitness. Therefore, understanding the temporal and contextual effects of NRF2 signaling is crucial for therapeutic design. The aim of this review is to examine how pharmacological modulation of NRF2 influences the invasive and metastatic dimensions of tumor progression, in addition to discussing its potential integration into TNM-based prognostic and treatment frameworks.

## 1. Introduction

Nuclear factor erythroid 2-related factor 2 (NRF2) is a transcription factor that orchestrates cellular defense mechanisms against oxidative and electrophilic stress by regulating the activity of antioxidant response elements (ARE). Its master regulator is the Kelch-like EC H-associated protein 1 (KEAP1) [1], which, under basal conditions, sequestrates NRF2 for proteasomal degradation; in response to oxidative stress or electrophiles, KEAP1 cysteine residues are modified, allowing NRF2 to accumulate and translocate to the nucleus to activate its target genes. This finely tuned system maintains the balance of redox homeostasis, promotes metabolic adaptation, and protects tissues from carcinogen-induced injury [2]. Importantly, NRF2 behaves as a “double-edged sword” in cancer biology: while its transient activation preserves genomic stability and prevents carcinogen-induced transformation during early stages of development, its persistent or constitutive activation—often driven by KEAP1, NFE2L2, or CUL3 alterations— supports oncogenesis in advanced tumors by promoting metabolic rewiring, epithelial–mesenchymal transition (EMT), invasion, and therapeutic resistance [3]. Conversely, persistent or constitutive activation—often due to mutations in KEAP1 or Nuclear factor erythroid 2-like 2 (NFE2L2) which encodes NRF2, or metabolic rewiring—confers a tumor-promoting phenotype, enhancing survival, proliferation, and treatment resistance in established cancers [4,5]. This chronic activation reprograms metabolism toward anabolic and antioxidant pathways, promotes epithelial–to–mesenchymal transition (EMT), and facilitates immune evasion and metastasis [6,7].

Recent clinical and experimental studies indicate that NRF2 activity correlates with distinct aspects of the tumor–node–metastasis (TNM) classification for malignant tumors, particularly with increased tumor invasion (T) and metastatic spread (M)—two dimensions strongly associated with poor patient prognosis. NRF2 signaling hyperactivity enhances cell motility, matrix degradation, and colonization capacity, whereas loss of NRF2 regulation promotes an aggressive and treatment-resistant phenotype [8]. Given these stage-dependent and context-specific functions, pharmacological modulation of NRF2 represents a promising but complex therapeutic avenue. In early tumorigenesis, NRF2 activators may act chemopreventively by strengthening antioxidant defenses, while in advanced disease, NRF2 inhibitors or metabolic interventions targeting NRF2-driven pathways could inhibit invasion and metastasis.

The objective of this review is, therefore, to analyze how NRF2 contributes mechanistically to the invasive (T) and metastatic (M) components of tumor progression and to evaluate how pharmacological modulation of this pathway (through direct inhibitors, activators, or metabolic reprogramming) could be exploited to interfere with these stages of cancer evolution.

### 1.1. NRF2 in Oxidative Stress, Disease, and Therapeutic Modulation

NRF2 is a master transcription factor that regulates the expression of antioxidant and detoxification genes through the KEAP1–NRF2–ARE pathway. Under basal conditions, NRF2 binds to its cytoplasmic repressor KEAP1, promoting CUL3-mediated ubiquitination and rapid proteasomal degradation. Upon exposure to oxidative or electrophilic stress, reactive cysteine residues in KEAP1 are modified, preventing NRF2 degradation and allowing its accumulation, nuclear translocation, and binding to AREs in target gene promoters [9,10]. Activated NRF2 induces a broad cytoprotective transcriptional program encompassing phase I and II detoxification enzymes, redox-balancing systems such as glutathione synthesis and NADPH regeneration, and metabolic regulators that maintain cellular redox balance. Through these pathways, NRF2 maintains redox balance, promotes metabolic homeostasis, and orchestrates adaptive responses to oxidative and electrophilic stress in normal tissues [11,12]. Physiologically, NRF2 functions as a key defender of tissue integrity, limiting the accumulation of reactive oxygen species (ROS) and protecting against environmental insults such as carcinogens, pollutants, and inflammation. This cytoprotective capacity reduces oxidative DNA damage and prevents mutagenesis and chronic inflammatory signaling, acting as a barrier to tumor initiation [13,14]. Accordingly, moderate NRF2 activation has been shown to prevent carcinogen-induced transformation and promote cellular detoxification in various models of chemical and environmental stress. Thus, NRF2 plays a preventive role in carcinogenesis, where its transient activation preserves genomic stability and promotes normal cell survival under stress conditions [12].

### 1.2. The KEAP1–NRF2–ARE Pathway: Basic Regulatory Mechanisms

NRF2 is a cap’n-collar basic leucine zipper transcription factor organized into seven Neh (NRF2-ECH homology) domains that coordinate its stability and transcriptional control. The Neh2 region contains the DLG and ETGE motifs that bind the Kelch domain of KEAP1, positioning NRF2 for ubiquitination, while the Neh1 domain mediates DNA binding and heterodimerization with small Maf proteins [9,15]. Structural analyses reveal a bipartite “hinge and latch” interaction that determines cytoplasmic sequestration of NRF2, where mutations in either motif disrupt repression and promote constitutive activation [16,17]. These modular domains integrate multiple stress-sensing inputs that collectively define the dynamic range of NRF2 activity [18].

Under basal conditions, KEAP1 functions as an adapter within the Cullin-3 (CUL3)–Ring box protein 1 (Rbx1) E3 ubiquitin ligase complex that continuously targets NRF2 for proteasomal degradation [19,20,21]. This process depends on double-site recognition of the DLG and ETGE motifs, allowing high-affinity binding and efficient ubiquitination cycles [16]. The fine-tuned turnover ensures low basal levels of NRF2, maintaining redox homeostasis without excessive expression of antioxidant genes. This constitutive degradation mechanism positions KEAP1 as a physical anchor and molecular rheostat for cytoplasmic NRF2 [19]. Upon exposure to oxidative or electrophilic stress, reactive cysteine residues on KEAP1—particularly Cys151, Cys273, and Cys288—undergo covalent modification that disrupts CUL3-dependent ubiquitination [22,23,24]. These cysteine adducts act as redox sensors, converting chemical stress into conformational changes that release NRF2, which translocates to the nucleus [25,26] where it heterodimerizes with small Maf proteins, and binds to AREs to drive transcription of cytoprotective genes involved in detoxification, glutathione synthesis, and NADPH regeneration [10].

### 1.3. NRF2 Pathogenic Activation in Cancer

Genetic alterations are a major driver of NRF2 dysregulation in cancer, particularly mutations in KEAP1, NFE2L2 (NRF2), and CUL3, which disrupt KEAP1-mediated ubiquitination and lead to constitutive NRF2 stabilization [27,28]. Mutations in the ETGE and DLG motifs of the Kelch domain are the most frequently mutated sites found in tumors [29]. Such mutations have been identified across multiple malignancies, including lung, esophageal, and renal carcinomas, where they correlate with aggressive phenotypes and therapy resistance [30,31]. In KEAP1- or CUL3-mutant tumors, the loss of functional repression enables continuous nuclear accumulation of NRF2, promoting a transcriptional program favorable to survival and proliferation [28]. The molecular transition from physiological NRF2 regulation to its pathogenic activation in cancer is illustrated in Figure 1, which contrasts the normal KEAP1-dependent ubiquitination and transient antioxidant response with the sustained, mutation-driven NRF2 hyperactivation observed in advanced tumors. Under homeostatic conditions, KEAP1 binds NRF2 through its DLG and ETGE motifs, enabling CUL3-mediated ubiquitination and proteasomal degradation. However, in the oncogenic context, KEAP1 cysteine modification, loss-of-function mutations, or p62 sequestration prevent NRF2 ubiquitination, leading to persistent nuclear accumulation and transcriptional activation of pro-survival, metabolic, and invasive genes.

### 1.4. Epigenetic and Non-Genetic Mechanisms of NRF2 Activation

Beyond genetic insults, NRF2 activation is also influenced by epigenetic mechanisms such as KEAP1 promoter hypermethylation and histone modifications that reduce KEAP1 expression [2,27]. Furthermore, the presence of competing proteins that bind KEAP1 can sterically inhibit NRF2 degradation, further stabilizing its activity in a mutation-independent manner [6]. These regulatory disruptions contribute to sustained NRF2 activation even in the absence of canonical oxidative stress signals [32].

### 1.5. Oncogenic and Metabolic Signaling Supporting NRF2 Dysregulation

Metabolic and oncogenic signaling pathways also play an important role in NRF2 dysregulation. Oncogenes such as KRAS, PI3K/AKT, and MYC have been shown to enhance NRF2 transcription and stability, creating a feedback loop that favors metabolic reprogramming under conditions that promote tumor proliferation.

Refs. [13,33] looked at tumor microenvironmental stressors, such as hypoxia and nutrient deprivation, which further activate NRF2 to maintain redox balance and anabolic metabolism [31]. This oncogenic metabolic integration strengthens cellular resilience and contributes to tumor progression [13]. NRF2 activity is also modulated at the post-translational level by phosphorylation and acetylation events that influence its nuclear translocation, DNA-binding affinity, and degradation [6,27]. Disruption of canonical KEAP1 interactions by alternative proteins or stress-responsive signaling cascades, such as endoplasmic reticulum stress pathways, may further contribute to non-canonical NRF2 activation [31]. These mechanisms, together, allow NRF2 stabilization beyond oxidative stress responses, promoting sustained transcriptional activation in advanced tumors [30].

### 1.6. Metabolic Reprogramming Driven by Persistent NRF2 Activation

Persistent NRF2 activation reprograms cellular metabolism toward antioxidant and anabolic pathways that promote tumor survival and proliferation [28,34]. This change enhances glutaminolysis and the pentose phosphate pathway, increasing NADPH generation and facilitating redox homeostasis under oncogenic stress conditions [13]. This reprogramming results in resistance to apoptosis, increased stem cell capacity, and adaptation to oxidative microenvironments, which favor tumor aggressiveness and therapeutic resistance [10,30].

Together, these pathogenic alterations support the conceptualization of NRF2 as a context-dependent oncogene. While transient activation provides chemopreventive effects in early carcinogenesis by reducing DNA damage and mutagenesis, constitutive hyperactivation in established tumors promotes progression, invasion, and drug resistance [2,28]. This dual function underscores the importance of temporal and spatial regulation of NRF2 activity in determining its protective or pathogenic effects in cancer [31].

## 2. NRF2 Crosstalk with Oncogenic and Inflammatory Pathways

NRF2 activity is profoundly shaped by its interaction with major oncogenic signaling pathways that support tumor survival and invasion. Among these, the PI3K/AKT axis is one of the most influential: activation of AKT inhibits GSK-3β, a kinase that normally phosphorylates NRF2 to promote β-TrCP-mediated degradation. By suppressing this degradation route, PI3K/AKT signaling stabilizes NRF2 and extends its nuclear retention, thereby enhancing antioxidant capacity, metabolic rewiring, and therapeutic resistance [5,35]. Similarly, MAPK/ERK signaling increases NRF2 transcriptional output during oxidative stress, reinforcing cytoprotective gene expression and promoting redox tolerance in malignant cells [13,33].

### 2.1. NRF2–TGF-β Interactions During EMT

TGF-β/Smad signaling also integrates tightly with the KEAP1–NRF2 axis. During epithelial–mesenchymal transition (EMT), TGF-β enhances Smad2/3 activation, which cooperates with NRF2 to reinforce hybrid epithelial/mesenchymal states that favor migration and invasion. Persistent TGF-β stimulation increases NRF2 accumulation, leading to elevated expression of genes involved in ECM remodeling, redox buffering, and metabolic adaptation [36]. This creates a positive feedback loop in which NRF2 supports EMT transcriptional programs while TGF-β signaling maintains conditions—oxidative stress, cytokine exposure, metabolic stress—that promote NRF2 hyperactivation. This cooperative interaction links redox adaptation directly to phenotypic plasticity, a hallmark of metastatic progression.

### 2.2. NRF2–NF-κB Crosstalk and Immunometabolic Regulation

NRF2 also exhibits reciprocal regulation with NF-κB, one of the central inflammatory pathways in cancer. Oxidative or inflammatory cues that activate NF-κB can simultaneously induce NRF2 transcription, while NRF2 in turn modulates the expression of cytokines, detoxifying enzymes, and metabolic regulators that influence NF-κB activity [6,13]. This crosstalk creates an immunometabolic environment favorable to tumor growth by suppressing immunogenic ROS signaling and maintaining chronic inflammation. Collectively, these interconnected pathways illustrate that NRF2 functions as an adaptive signaling hub, integrating oncogenic, metabolic, and inflammatory cues. This interconnected regulatory architecture underscores why therapeutic modulation of NRF2 requires a coordinated, systems-level strategy rather than isolated targeting of a single pathway.

## 3. Invasion Coupled with Metastasis

The functional identity of NRF2 in cancer is profoundly dualistic. In normal tissues and during early carcinogenesis, transient NRF2 activation preserves redox homeostasis, promotes detoxification, and prevents oxidative DNA damage, thereby exerting a tumor suppressor effect. However, in advanced or genetically altered tumors, persistent NRF2 hyperactivation—driven by mutations in KEAP1 or NFE2L2, oncogenic PI3K/AKT or KRAS signaling, and metabolic stress—transforms this protective program into a protumoral mechanism. Constitutive NRF2 activity enhances metabolic plasticity, epithelial–mesenchymal transition, angiogenesis, and immune evasion, promoting invasion and metastatic spread. This paradox establishes NRF2 as a molecular switch whose impact on cancer progression depends on timing, severity, and cellular context.

### 3.1. NRF2 as a Regulator of Epithelial–Mesenchymal Transition (EMT)

NRF2 has emerged as a regulator of EMT, the developmental program that endows cancer cells with motility and pluripotency. Its activation stabilizes hybrid epithelial/mesenchymal states and promotes collective migration by maintaining partial EMT transcriptional networks such as SNAIL, ZEB1, and TWIST [37,38]. By modulating ROS and Notch signaling, NRF2 fine-tunes Snail expression and promotes pluripotency-like phenotypes that enhance metastatic competence [39,40]. In this way, the pathway intersects with canonical drivers of EMT while preserving redox homeostasis, creating a permissive environment for invasion [6,28,34].

### 3.2. ECM Remodeling and Cytoskeletal Dynamics

Beyond transcriptional control of EMT, NRF2 orchestrates cytoskeletal remodeling and extracellular-matrix (ECM) degradation, enabling local tissue invasion. Up-regulation of matrix metalloproteinases such as MMP-2 and MMP-9 and activation of integrin signaling promote renewal of adhesion and motility in NRF2-elevated tumors [6,41,42]. NRF2-driven redox signaling also regulates collagen and ECM-remodeling genes, linking antioxidant defense to the physical dynamics of tumor dissemination [43]. These findings are consistent with broader frameworks positioning NRF2 among the key drivers of cancer invasion and migration [2,44].

### 3.3. Metabolic Reprogramming Supporting Invasion and Metastasis

A defining feature of metastatic progression is metabolic flexibility, and NRF2 plays a pivotal role by rewiring glucose and glutamine metabolism to maintain biosynthesis and redox balance. Constitutive NRF2 activation enhances flux through the pentose-phosphate pathway (PPP) and glutaminolysis, providing NADPH and anabolic precursors for proliferative and invasive growth [8,13]. The NRF2–Coactivator associated arginine methyltransferase 1 (CARM1) axis epigenetically regulates PPP enzymes in gastric cancer, whereas MYC-dependent NRF2 activation drives similar programs in head and neck tumors [45,46]. NRF2-dependent cancers rely on this metabolic adaptation to survive under oxidative and nutrient stress [47,48].

### 3.4. Immune Evasion and Metastatic Colonization

NRF2 also contributes to immune evasion and oxidative-stress resistance that promote metastatic colonization. It directly induces PD-L1 transcription, attenuating antitumor immunity and influencing responses to PD-1 blockade [36,49,50]. By suppressing immunogenic ROS signaling and reshaping cytokine profiles, NRF2 hyperactivation generates “immune-cold” tumor microenvironments [51,52]. Clinically, KEAP1 or NFE2L2 mutations correlate with poor immunotherapy outcomes and decreased T-cell infiltration [45,53], underscoring the dual role of NRF2 in redox control and immune evasion.

### 3.5. Clinical Implications and Integration into TNM Progression

Together, the phenomena mentioned previously form the molecular framework by which NRF2 drives tumor invasion (T) and metastasis (M) within the TNM Classification of Malignant Tumors system. Elevated NRF2 expression or KEAP1/NFE2L2 mutations are associated with advanced pathological stages, higher tumor burden, and worse prognosis in multiple cancers [5,54,55,56] as demonstrated by extensive clinical evidence, summarized in Table 1. Therefore, integrating NRF2 status into TNM-based models can refine prognostic assessment and identify patients who might benefit from NRF2-targeted therapeutic strategies [2].

### 3.6. Pharmacological Modulation of NRF2: Mechanisms and Effects

#### 3.6.1. Mechanistic Basis of Pharmacologic Intervention

Pharmacological modulation of NRF2 can occur at multiple regulatory levels, ranging from transcriptional control to post-translational modifications and signaling interactions. At the transcriptional and epigenetic level, drugs such as the DNA-methyltransferase inhibitor azacitidine and the histone deacetylases (HDAC) inhibitor vorinostat can restore KEAP1 expression by reversing promoter hypermethylation, thereby reducing constitutive NRF2 activation [77,78]. In contrast, the bromodomain inhibitor JQ1 downregulates NFE2L2 transcription by disrupting bromodomain-containing protein 4 (BRD4)-dependent chromatin activation, attenuating NRF2 signaling [79]. These approaches highlight how the NRF2 pathway is epigenetically “tunable,” offering opportunities for therapeutic reprogramming, although clinical studies directly targeting this axis remain [80]. At the post-translational level, NRF2 activity is tightly regulated through phosphorylation, acetylation, and ubiquitination. Kinase inhibitors targeting PI3K/AKT (LY294002) or ERK (U0126) pathways can decrease NRF2 phosphorylation and nuclear stability, thereby reducing the expression of antioxidant genes [35,81]. Similarly, AMPK activators such as metformin and AICAR suppress anabolic NRF2 signaling and restore redox sensitivity in KEAP1-mutant tumors [5]. These findings support the concept that manipulating upstream kinase networks can recalibrate NRF2 oncogenic activity. Early-phase clinical trials evaluating buparlisib (a PI3K inhibitor) in solid tumors provide translational relevance to this approach, demonstrating a reduction in metabolic resilience in cancers with highly active NRF2 [35]. Interference at the level of protein–protein interaction represents another mechanistic pathway. Small molecules such as brusatol, ML385, and halofuginone directly disrupt the KEAP1–NRF2–CUL3 complex, accelerating NRF2 degradation or preventing its binding to DNA [82,83]. Brusatol promotes proteasomal elimination of NRF2 independently of KEAP1, reducing the expression of cytoprotective enzymes and sensitizing tumors to chemotherapy. Halofuginone, by inhibiting prolyl-tRNA synthetase and the TGF-β/Smad axis, indirectly decreases NRF2-induced VEGF and MMP expression, limiting invasion and angiogenesis [5]. Although still in preclinical phase, these inhibitors demonstrate the feasibility of the concept for the direct suppression of NRF2′s oncogenic functions. Pharmacological intervention of this pathway can also operate through metabolic feedback mechanisms that alter the redox balance. Buthionine sulfoximine (BSO) depletes glutathione, sensitizing cells to oxidative stress, while N-acetylcysteine (NAC) restores intracellular thiols and modulates the reactivity of KEAP1 cysteine [26]. By influencing the oxidation states of cysteine, these agents modify the balance between NRF2 degradation and stabilization, effectively acting as redox-sensitive switches. Such interventions have shown promising results in preclinical models of redox-addicted cancers, though clinical validation remains limited. Finally, the interaction between signaling pathways links NRF2 activity to broader oncogenic networks. Pathways such as TGF-β/Smad, MAPK/ERK, NF-κB, and mechanistic target of rapamycin (mTOR) modulate the transcriptional expression of NRF2, either enhancing or repressing its activity depending on the cellular context [5,35]. For example, inhibition of mTOR with rapamycin can decrease NRF2 nuclear translocation, whereas activation of NF-κB increases its transcriptional persistence under stress. These convergent pathways underscore why NRF2 cannot be considered a linear signaling cascade, but rather as an integrative node within the tumor’s adaptive network. Overall, these findings illustrate that NRF2 is pharmacologically amenable to being targeted through multiple molecular levels: transcriptional, post-translational, structural, metabolic, and signaling. Emerging trials combining PI3K/mTOR inhibitors or glutaminase blockers (CB-839) with conventional chemotherapy in KEAP1-mutant non-small-cell lung cancer exemplify how mechanistic insights are being translated into clinical strategies [84].

The growing recognition of NRF2′s dual role—as a guardian against oxidative damage and as a facilitator of malignant progression—creates both a biological paradox and a therapeutic opportunity. Having established how persistent NRF2 activation promotes epithelial–mesenchymal transition, metabolic reprogramming, and immune evasion, the next logical step is to explore how this pathway can be selectively manipulated. Pharmacological modulation of NRF2 aims to rebalance this deregulated network by enhancing its protective functions during early carcinogenesis or by suppressing its oncogenic output in advanced disease. Understanding the molecular layers at which NRF2 can be targeted—transcriptional, post-translational, metabolic, and signaling—provides a framework for rational drug design and for integrating NRF2 status into personalized treatment strategies. An overview of the main pharmacological agents targeting the KEAP1–NRF2 axis, including their mechanisms of action and experimental doses, is summarized in Table 2.

#### 3.6.2. NRF2 Inhibitors and Their Anti-Invasive Mechanisms

Pharmacological inhibition of NRF2 has emerged as a promising approach to overcome chemoresistance and suppress invasion in NRF2-dependent tumors. Brusatol is one of the earliest identified NRF2 inhibitors, accelerating proteasomal degradation of NRF2 independently of KEAP1 and leading to a rapid and transient depletion of downstream antioxidant proteins such as HO-1 and NAD(P)H quinone dehydrogenase 1 (NQO1) [85]. This transient inhibition reverses EMT characteristics, restoring E-cadherin expression while reducing vimentin, thereby sensitizing lung and pancreatic cancer cells to cisplatin and gemcitabine [83,98]. Although it is highly effective in preclinical studies, there are currently no active clinical trials evaluating brusatol in humans, largely due to concerns about off-target toxicity and poor pharmacokinetic stability.

Another well-characterized small molecule, ML385, directly targets the Neh1 DNA-binding domain of NRF2, preventing its association with small Maf proteins and the subsequent transcriptional activation of ARE-regulated genes [86]. This compound reduces NRF2-dependent antioxidant defenses and affects metastatic potential in non-small-cell lung carcinoma (NSCLC) and head and neck squamous carcinoma models [87]. ML385 has also been shown to have the ability to resensitize chemoresistant tumors to platinum-based therapies, suggesting a therapeutic synergy in KEAP1-deficient cancers [99]. Despite these preclinical successes, ML385 has not yet advanced to clinical trials, and ongoing research seeks to improve its bioavailability and specificity before human studies are initiated. Halofuginone, a derivative of febrifugine, exhibits a dual inhibitory action by suppressing both the TGF-β/Smad2/3 signaling axis and NRF2 accumulation [88]. Through this mechanism, it down-regulates VEGF, HIF-1α, and MMP-9, thereby attenuating angiogenesis and invasion in tumors with high NRF2 expression. The antitumor efficacy of halofuginone has been demonstrated in models of lung adenocarcinoma and melanoma, where nanoparticle formulations significantly reduced systemic toxicity while maintaining NRF2 inhibition [89]. It is important to highlight that halofuginone has been evaluated in early-phase clinical trials: evaluation—[Phase 1 (NCT00027677)] in advanced solid tumors and Phase 2 [(NCT00064142)] in Kaposi’s sarcoma, establishing human tolerability and evidence of safety, although NRF2 suppression was not directly assessed in those studies.

Polyphenols such as trigonelline and luteolin have also been identified as NRF2 suppressors. These compounds restore KEAP1 expression by demethylating its promoter or inhibiting PI3K/AKT signaling, which reduces NRF2 nuclear translocation [100,101,102]. In prostate and gastric cancer models, luteolin decreases NRF2-dependent antioxidant capacity, increasing ROS accumulation and promoting apoptosis in metastatic cells. Although clinical validation is lacking, several ongoing observational studies on dietary flavonoids are evaluating their potential to modulate oxidative and inflammatory biomarkers in cancer prevention (ClinicalTrials.gov search, 2025). Cardamonin, a chalcone present in *Alpinia* species, has been described as a paradoxical modulator of NRF2. At low doses, cardamonin covalently modifies reactive cysteine residues in KEAP1, hindering NRF2 transactivation and increasing ROS generation. At higher concentrations, it can activate cytoprotective genes [103,104]. Through this redox-dependent modulation, cardamonin suppresses tumor migration and enhances TRAIL-induced apoptosis in preclinical models. Despite its promising mechanistic profile, no registered interventional trials have yet tested cardamonin as an NRF2 inhibitor in human cancers.

Collectively, these NRF2 inhibitors act through complementary mechanisms—enhancing proteasomal degradation (brusatol), blocking DNA binding (ML385), interfering with pro-oncogenic signaling (halofuginone), or restoring KEAP1 expression (trigonelline, luteolin, cardamonin). By re-establishing ROS-mediated cytotoxicity and reversing EMT transcriptional programs, they attenuate invasion, metastasis, and drug resistance in NRF2-driven malignancies. Continued refinement of their pharmacokinetics and selectivity, together with insights from early clinical studies such as the halofuginone trials, may ultimately enable their translation into targeted anti-metastatic therapies.

### 3.7. NRF2 Activators with Chemopreventive Potential

Electrophilic NRF2 activators are among the most studied chemopreventive agents, acting by covalent modification of KEAP1 cysteine residues. Sulforaphane, a dietary isothiocyanate derived from cruciferous vegetables, alkylates critical cysteines such as Cys151 in KEAP1, leading to transient stabilization of NRF2 and the induction of detoxifying enzymes such as HO-1 and NQO1 [90,91]. Similarly, the synthetic triterpenoid bardoxolone methyl (CDDO-Me) modifies multiple cysteines of KEAP1 and promotes NRF2 activation, reducing oxidative DNA damage and fibrosis in models of early carcinogenesis. Dimethyl fumarate (DMF), an electrophilic ester of fumaric acid, activates NRF2 by modifying Cys151 and Cys273 and has demonstrated chemopreventive effects in colon and skin cancer models, in addition to its clinical validation in multiple sclerosis [94].

In parallel, several indirect activators enhance NRF2 activity through upstream kinase signaling or epigenetic modulation. Polyphenols such as resveratrol activate AMPK and p38 MAPK, promoting NRF2 nuclear translocation and transcriptional output, associated with cytoprotection and redox homeostasis [105,106]. Curcumin has been shown to induce NRF2 activation through inhibition of HDACs, enhancing NRF2 acetylation and prolonging its nuclear retention [107,108]. These compounds function as indirect NRF2 enhancers by altering the cellular redox balance or modifying chromatin accessibility, thereby priming antioxidant and xenobiotic defenses. Mixtures of natural compounds such as epigallocatechin gallate (EGCG) have also demonstrated chemopreventive effects in early stages by inducing NRF2. It enhances NRF2-ARE signaling to increase antioxidant resistance during the initiation phase of carcinogenesis, reducing ROS-mediated mutagenesis without promoting hyperproliferation in normal tissues [109]. By transiently increasing the expression of detoxifying enzymes, these natural agents contribute to the neutralization of carcinogens and the reduction in genomic instability associated with inflammation [108]. From a mechanistic perspective, electrophilic and indirect activators converge on increased antioxidant capacity, glutathione synthesis, and xenobiotic clearance, thereby reducing DNA damage and carcinogen-induced transformation during early tumorigenesis [110,111]. This transient activation of NRF2 promotes cellular defense and immune homeostasis, preserving genomic integrity in premalignant contexts [112]. This cytoprotective effect explains the clinical and experimental success of dietary and pharmacological NRF2 inducers in cancer prevention strategies. However, sustained and dysregulated NRF2 activation in established tumors can lead to metabolic reprogramming, therapy resistance, and survival advantages, raising concerns about the prolonged use of activators in advanced-stage malignancies [2,30]. Therefore, while NRF2 activators have significant chemopreventive potential when administered at appropriate stages, their therapeutic application requires precise timing to avoid promoting tumor progression in NRF2-dependent cancers [31].

#### 3.7.1. Targeting NRF2-Dependent Metabolic Pathways

Constitutive activation of KEAP1/NRF2 reprograms tumor metabolism toward glutaminolysis and PPP flux, generating NADPH and GSH to buffer oxidative stress and promote invasive growth [5,13]. In KRAS-driven lung cancer, Keap1 loss creates a glutaminase dependency that can be therapeutically exploited with CB-839/telaglenastat [113]. The cooperation between LKB1 and KEAP1/NRF2 further increases glutamine dependency and sensitizes tumors to glutaminase inhibition in vitro and in vivo [114]. These data provide a mechanistic basis for evaluating glutaminase blockade in cancers with alteration in KEAP1/NFE2L2. The KEAPSAKE trial [NCT04265534] is a randomized phase 2 study evaluating telaglenastat + pembrolizumab + chemotherapy versus placebo as first-line treatment for stage IV non-squamous NSCLC with KEAP1/NRF2 mutation, with progression-free survival (PFS) and safety as the primary endpoints [84], but it was terminated before its contemplated completion date because of the lack of clinical benefit. A phase 1 study combining sapanisertib (TORC1/2 inhibitor) + CB-839 to determine recommended dose and preliminary efficacy in advanced NSCLC, focusing on NFE2L2/KEAP1 subgroups [115]. Taken together, these trials highlight the metabolic vulnerability of NRF2-dependent tumors. Inhibition of the PPP also undermines NRF2-mediated redox resilience. Glucose-6-phosphate dehydrogenase (G6PD) is a key regulator of NADPH production; overexpression of G6PD/TKT downstream of NRF2–MYC drives nucleotide biosynthesis and progression in head and neck squamous cell carcinoma (HNSCC) [116]. Inhibiting G6PD decreases NADPH production, induces immunogenic cell death, and can potentiate immunotherapies, suggesting synergy in tumors with high NRF2 expression [95]. Several reviews further solidify the importance of the pentose phosphate pathway in tumor redox control and its therapeutic potential [117]. mTOR/PI3K signaling interacts with NRF2 via autophagy and p62. mTORC1-dependent phosphorylation of p62 promotes KEAP1 sequestration, stabilizing NRF2; therefore, mTOR/PI3K inhibition can reduce this non-canonical activation and decrease NRF2 nuclear activity [118]. The autophagy literature shows competition between p62 and KEAP1 as a key amplifier of NRF2 signaling under stress, linking nutrient sensing to redox homeostasis [119,120]. These mechanisms justify combining PI3K/mTOR blockers with NRF2-pathway inhibition to mitigate invasion and resistance to therapy.

Finally, AMPK acts as a metabolic brake on the anabolic programs that maintain metastatic capacity. While the interaction between AMPK and NRF2 is context dependent, pharmacological activation of AMPK can limit growth signals and sensitize tumors: metformin has been shown to suppress NRF2-mediated chemoresistance and attenuate HO-1/antioxidant responses in cancer models [83,96]. In theory, combining AMPK activators with PPP or glutaminase inhibitors could deplete reducing equivalents and expose NRF2-activated tumor subtypes to oxidative damage.

In summary, metabolic interventions—glutaminase blockade, PPP inhibition, PI3K/mTOR pathway suppression, and AMPK activation—exploit redox vulnerabilities generated by constitutive NRF2 activation, allowing invasive cancers to become more sensitive to oxidative stress and conventional therapy [5,13]. Ongoing clinical trials with telaglenastat (alone or with mTOR inhibitors) exemplify how these mechanistic insights are being translated into patient-targeted strategies.

#### 3.7.2. Interconnected Regulatory Networks in NRF2 Pharmacological Control

Pharmacological modulation of NRF2 operates within a complex and highly interconnected signaling network that integrates redox, metabolic, and immune regulatory signals. Upstream kinase pathways, including PI3K/AKT, MAPK/ERK, and TGF-β/Smad, dynamically regulate NRF2 phosphorylation, stability, and nuclear import [121]. Activation of the PI3K/AKT pathway stabilizes NRF2 by inhibiting GSK-3β-mediated degradation, while MAPK signaling promotes its transcriptional capacity under stress conditions [5]. In parallel, TGF-β/Smad signaling contributes to NRF2 accumulation and the progression of the epithelial–mesenchymal transition (EMT), establishing a positive feedback loop that links oxidative adaptation with invasive behavior [36]. These ascending cascades demonstrate that NRF2 activity is not autonomous, but rather orchestrated through canonical oncogenic pathways.

At the level of central regulation, direct covalent modification or binding of small molecules to the cysteine residues of KEAP1 represents a fundamental pharmacological control mechanism. Electrophilic compounds, such as sulforaphane or bardoxolone methyl alkylate, alkylate the reactive cysteines aforementioned on KEAP1, preventing CUL3-dependent ubiquitination and promoting NRF2 stabilization [5]. Conversely, inhibitors such as brusatol or halofuginone interfere with the KEAP1–NRF2–CUL3 complex, restoring proteasomal degradation and redox vulnerability [83,88]. The delicate balance between activation and inhibition at this point underscores the dual therapeutic challenge: transient activation favors chemoprevention, while chronic activation promotes tumor persistence. Subsequent feedback mechanisms further modulate NRF2 activity through metabolic and autophagic cross-regulation. The adaptor protein p62/SQSTM1 competes with NRF2 for binding to KEAP1; its phosphorylation at Ser349 disrupts the KEAP1–CUL3 complex, leading to sustained NRF2 activation during oxidative or metabolic stress [122]. This p62–KEAP1 axis acts as an autophagy-dependent amplifier, enabling tumor cells to couple proteostasis and antioxidant defense [123]. Metabolically, NRF2-driven expression of enzymes from the PPP and glutaminolysis generates NADPH and glutathione, providing the reducing equivalents necessary for invasive growth.

On a systemic level, NRF2 interacts with immunometabolic and therapeutic networks that determine tumor progression and therapy response. Constitutive activation of NRF2 increases PD-L1 expression and suppresses immunogenic ROS signaling, contributing to immune evasion [50,89]. Combination strategies integrating NRF2 inhibitors with PD-1/PD-L1 blockade or ROS-generating chemotherapeutics seek to exploit this vulnerability by restoring oxidative stress in the tumor microenvironment [36]. Similarly, simultaneous inhibition of NRF2 with metabolic or autophagic modulators offers a way to sensitize tumors resistant to standard therapy. These multidimensional interactions position NRF2 as a regulatory hub. Its modulation affects the redox, metabolic, and immune environments, determining the invasive and metastatic potential. Taken together, this integrated view underscores that NRF2 signaling cannot be isolated to a single molecular pathway. Its pharmacological modulation requires a systemic perspective that considers upstream oncogenic factors, central redox regulators, downstream metabolic feedbacks, and the broader immune context. Recognizing this interconnectedness is essential for designing stage-specific interventions that suppress pro-tumor functions of NRF2 while preserving its physiological cytoprotective roles.

## 4. Limitations

One of the principal limitations identified in this review is the heterogeneity of evidence surrounding NRF2′s role across tumor types and disease stages. Although its activation has been consistently associated with chemoresistance, invasion, and metastasis, the context-dependent behavior of NRF2 makes it difficult to generalize findings. As noted by Occhiuto et al. [4], NRF2 may function as a tumor suppressor in the early phases of carcinogenesis but as an oncogenic driver in advanced stages. This duality complicates both experimental design and therapeutic targeting, as the same pathway can produce opposing effects depending on cellular context, mutation profile, and tumor microenvironmental conditions.

A second limitation is the limited translational evidence supporting pharmacologic modulation of NRF2. Despite extensive preclinical data linking KEAP1/NFE2L2 mutations to poor prognosis, clinical trials have produced inconsistent or null results. The termination of the KEAPSAKE trial evaluating telaglenastat [124] illustrates the challenge of translating molecular rationale into clinical benefit. Similarly, the ongoing sapanisertib trial (NCT05275673) has yet to demonstrate efficacy, emphasizing the gap between mechanistic understanding and clinical validation. These outcomes suggest that NRF2-targeted interventions may require better patient selection and combination strategies rather than monotherapy approaches.

Another major limitation lies in the lack of standardized biomarkers for NRF2 activity and localization. Chen et al. [119] demonstrated that cytoplasmic, rather than nuclear, NRF2 expression predicts poor chemotherapy response, yet such findings are not consistently replicated or applied in clinical settings. Moreover, as Ricciuti et al. [53] and Zhu et al. [124] described, the coexistence of KEAP1 and STK11 mutations confers immune resistance and poor outcomes, but the field lacks harmonized diagnostic frameworks to stratify patients according to NRF2 dependency.

The absence of uniform biomarkers limits the reproducibility of studies and hinders the development of precision-based NRF2 therapeutics. From a methodological standpoint, many studies rely heavily on in vitro or animal models, which may not accurately capture the complexity of NRF2 signaling in human cancers. Rojo de la Vega et al. [2] emphasized that NRF2 interacts with multiple hallmarks of cancer through intricate feedback networks, suggesting that simplified experimental systems may overlook critical crosstalk mechanisms. Furthermore, variations in assay design, cell line genetics, and oxidative stress conditions contribute to inconsistent results, making cross-study comparisons difficult. This experimental variability reinforces the need for systems biology and multi-omic approaches to validate NRF2′s functional relevance across diverse contexts. Finally, the therapeutic window for NRF2 modulation remains narrow, as highlighted by Lin et al. [125].

Systemic inhibition risks compromising NRF2′s essential physiological roles in redox balance and cytoprotection, whereas overactivation may inadvertently support tumor adaptation. The ubiquitous expression of NRF2 across normal tissues makes it challenging to achieve selective inhibition without toxicity. Consequently, future efforts should focus on developing context-specific delivery systems, such as nanoparticle-based or tissue-selective inhibitors, alongside biomarker-guided trial designs to minimize off-target effects and optimize therapeutic precision.

### Discussion and Future Perspectives

The growing understanding of NRF2′s role throughout the cancer process underscores the need for selective and context-dependent modulation, rather than global activation or inhibition. As Occhiuto et al. [4] point out, the KEAP1–NRF2 axis acts as a tumor suppressor in the early stages of carcinogenesis but becomes oncogenic in later stages, depending on the tumor type and its microenvironment. This duality demands precision in targeted therapy, aiming to preserve NRF2′s cytoprotective function in healthy tissues while avoiding the chronic activation that promotes tumor survival, invasion, and treatment resistance. Although there is ample evidence correlating high NRF2 expression with poor prognosis and therapeutic resistance, clinical interventions directly targeting this pathway remain scarce. The challenge lies in NRF2′s ubiquitous tissue distribution and essential physiological functions, which make its pharmacological inhibition difficult. In colorectal cancer, NRF2 overexpression has been linked to distant metastasis through the induction of HO-1 [126], while elevated KEAP1 levels in tumor tissue have been associated with lymphovascular invasion [127]. Similar patterns have been described in non-small-cell lung cancer, where high NRF2 expression correlates with lymph node or distant metastasis [54], as well as in esophageal squamous cell carcinoma [46], osteosarcoma [128], and papillary renal cell carcinoma [129]. These consistent associations suggest that NRF2 expression could complement the TNM classification as a practical biomarker for identifying aggressive disease and anticipating treatment resistance. If NRF2 were to be considered a therapeutic target, the most logical approach would be to evaluate its inhibition in advanced stages of the disease, particularly in those with lymph node invasion or metastasis, where its oncogenic role is most pronounced. Several clinical studies have begun exploring this translational approach. A phase II trial was initiated with telaglenastat, a glutaminase inhibitor designed for tumors with mutations in the KEAP1/NRF2 pathway (NCT03872427), and another trial was initiated targeting stage IV non-small-cell lung cancer (NCT04265534); however, the latter was discontinued in 2022 after failing to demonstrate clinical benefit. Additionally, an ongoing phase II trial is evaluating sapanisertib, a dual mTORC1/2 inhibitor, in patients with relapsed or refractory NRF2-mutant NSCLC (NCT05275673). In parallel, Peng et al. [130] proposed the use of NRF2 modulators in Barrett’s esophagus, arguing that the progressive increase in ROS and NRF2 activity during malignant transformation could be exploited therapeutically—either through activation in early stages or inhibition once invasive potential emerges. These examples highlight both the promise and complexity of translating NRF2 modulation into clinical benefits. Future progress will likely depend on rational combination strategies that integrate NRF2 modulation with immunotherapy and metabolic interventions. Ricciuti et al. [53] demonstrated that KEAP1 and STK11 mutations generate an immune-exclusionary tumor microenvironment, limiting PD-(L)1 response in NSCLC; therefore, combining metabolic or redox modulation with immune checkpoint blockade could restore sensitivity.

Zhu et al. [124] similarly demonstrated that KEAP1/NFE2L2 mutations predict poor outcomes with both chemotherapy and immunotherapy, reinforcing the rationale for dual-track interventions that disrupt the redox–immune resistance axis. The design of biomarker-guided clinical trials is fundamental to realizing this vision. Chen et al. [119] discovered that cytoplasmic, rather than nuclear, NRF2 localization correlates with lower survival and reduced platinum response, highlighting a potential morphological marker for identifying NRF2-dependent tumors. Integrating genetic (KEAP1/NFE2L2/STK11), transcriptional, and metabolic profiles will allow for patient stratification and optimize therapy selection. Finally, a systems biology framework should underpin future efforts. Rojo de la Vega et al. [2] demonstrated that NRF2 regulates multiple distinctive cancer characteristics, such as proliferation, angiogenesis, metabolism, and metastasis, through complex feedback loops. Computational modeling of these networks can identify context-specific vulnerabilities, enabling more targeted interventions. Complementarily, Lin et al. [125] caution that indiscriminate antioxidant therapies could paradoxically promote tumor survival by maintaining NRF2-mediated redox balance.

In summary, the future of NRF2-targeted therapy will depend on integrating molecular knowledge with clinical stratification. Through selective modulation, combination regimens, and systems-level modeling, NRF2 could be transformed from a paradoxical challenge into a predictive and actionable target within precision oncology.

## 5. Concluding Remarks

NRF2 represents a molecular paradox, as it acts simultaneously serving as a guardian of cellular homeostasis and as an inducer of malignant progression. Its protective role in detoxification and oxidative balance contrasts sharply with its ability to maintain tumor survival, invasion, and therapeutic resistance when persistently activated. This duality places NRF2 at the crossroads between cancer prevention and cancer promotion, demanding a nuanced understanding of its biological context. The evidence highlights that temporal and contextual precision is essential in the development of therapies. Strategies targeting transient activation may be beneficial for preventing carcinogenesis or reducing tissue damage, while selective inhibition could counteract the pro-tumor effects of NRF2 in advanced disease. This differentiation requires the integration of molecular profiles, stage-specific biomarkers, and systemic approaches to guide intervention timing and patient selection. In short, pharmacological modulation of NRF2 offers a promising avenue for transforming this complex transcription factor from an oncogenic facilitator into a clinically useful target. By aligning therapeutic design with the molecular and temporal profile of each tumor, NRF2-targeted strategies could evolve into personalized treatments, harnessing its protective potential while neutralizing its malignant influence.

## Figures and Tables

**Figure 1 medicina-61-02224-f001:**
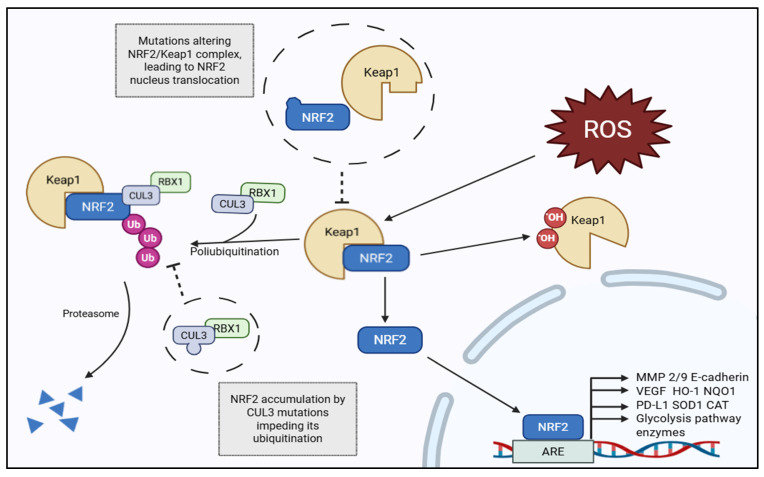
NRF2 Signaling is enhanced by an enhanced unbinding of NRF2 to KEAP1. Dysregulation of the KEAP1–NRF2 Axis in Oxidative Stress and Tumor Progression. During metastatic progression, sustained oxidative stress or mutations in KEAP1, NFE2L2, or CUL3 prevent NRF2 ubiquitination. Persistent NRF2 activation enhances metabolic reprogramming, epithelial–mesenchymal transition, matrix remodeling (MMP 2/9), and angiogenesis, driving an invasive mesenchymal phenotype characterized by therapy resistance and metastasis. Abbreviations: ARE, Antioxidant Response Elements; CUL3, Cullin 3; HO-1, Heme Oxygenase-1; KEAP1, Kelch-like ECH-associated protein 1; MMP, Matrix Metalloproteinase; NQO1, NAD(P)H Quinone Dehydrogenase 1; NRF2, Nuclear factor erythroid 2-related factor 2; PD-L1, Programmed Death-Ligand 1; RBX1, Ring Box Protein 1; Ub, Ubiquitin.

**Table 1 medicina-61-02224-t001:** Clinical relevance of NRF2/KEAP1 and CUL3 mutations in different types of cancer compared to non-mutated variants. pNRF2: phosphorylated NRF2; WT: wild type.

Mutation	Cancer	Clinical Outcome	Reference
NRF2	Adrenocortical carcinoma	Worse prognosis vs. WT	[57]
	Anaplastic glioblastoma	Worse prognosis vs. WT	[58]
	Anaplastic glioma	Worse prognosis vs. WT	[58]
	Esophageal squamous cell carcinoma	Worse prognosis vs. WT	[59]
	Esophageal squamous cell carcinoma	Chemo and radiotherapy resistance in advanced states, and worse prognosis vs. WT	[60]
	Esophageal squamous cell carcinoma	High expression associated with worse overall survival	[46]
	Hepatocarcinoma	Worse prognosis in low expression group and disease-free survival	[61]
	Lung adenocarcinoma	Worse prognosis and chemosensivity vs. WT	[62]
	Lung squamous cell carcinoma	Worse prognosis vs. WT	[63]
	Lung squamous cell carcinoma	Worse prognosis vs. WT	[59]
	Lung adenocarcinoma	No difference in prognosis vs. WT	[59]
	Lung adenocarcinoma	Lower disease-free survival vs. WT	[64]
	Non-small cell lung carcinoma	Lower overall survival vs. WT	[65]
	Melanoma	Worse prognosis vs. WT	[66]
	Melanoma	Worse prognosis vs. WT	[67]
	Renal cell carcinoma	Worse prognosis vs. WT	[68]
	Triple-negative breast cancer	Higher pNRF2 levels predicted worse response to neoadjuvant chemotherapy	[69]
KEAP1	Esophageal squamous carcinoma	No difference in prognosis vs. WT	[59]
	Lung adenocarcinoma	No difference in prognosis vs. WT	[59]
	Lung adenocarcinoma	Higher rate of mutated KEAP1 in lymph node and distant metastasis patients, lower overall survival vs. WT	[70]
	Lung adenocarcinoma	Lower progression-free survival and overall survival vs. WT	[71]
	Lung squamous cell carcinoma	Worse prognosis vs. WT	[59]
	Non-small cell lung carcinoma	Lower overall survival vs. WT	[65]
	Non-small cell lung carcinoma	Shorter time to treatment failure and worse overall survival vs. WT	[72]
	Renal cell carcinoma	No difference in prognosis vs. WT	[68]
CUL3	Head and neck squamous cell carcinoma	Worse prognosis vs. WT	[73]
	Head and neck squamous cell carcinoma	Worse prognosis vs. WT	[74]
	Glottic cell squamous cell carcinoma	Worse prognosis vs. WT	[75]
	Non-small cell lung carcinoma	Lower overall survival vs. WT	[76]

**Table 2 medicina-61-02224-t002:** Therapeutic Compounds Targeting NRF2: Mechanisms, Doses, and References. Pharmacological modulators of the KEAP1–NRF2 pathway, including activators, inhibitors, and metabolic regulators. Compounds are listed with their mechanism of action, experimental dose when available, and supporting references.

Drug	Dose	Mechanism of Action	Reference
Brusatol	Not standardized	Promotes NRF2 degradation; sensitizes tumors	[83,85]
ML385	Not standardized	Blocks NRF2 DNA binding	[86,87]
Halofuginone	Not standardized	Inhibits TGF-β/Smad; reduces NRF2 accumulation	[88,89]
Sulforaphane	5–20 μM	KEAP1 cysteine alkylation → NRF2 activation	[90,91]
CDDO-Me (bardoxolone)	10–100 nM	Electrophilic covalent activation of NRF2	[92,93]
Dimethyl fumarate	240 mg/day	Electrophilic KEAP1 modification	[94]
CB-839 (Telaglenastat)	Not standardized	Glutaminase inhibitor	[84]
G6PD inhibitors	Not standardized	Blocks PPP/NADPH production	[95]
PI3K/mTOR inhibitors	Not standardized	Reduces NRF2 stabilization	[35]
Metformin	250–2000 mg/day	AMPK activation suppresses NRF2 chemoresistance	[96,97]

## Data Availability

Not applicable.

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
