# Peer review of "Effects of the Pharmacological Modulation of NRF2 in Cancer Progression"

_medicina, 2025, doi:10.3390/medicina61122224_

Round 1
Reviewer 1 Report
Comments and Suggestions for Authors
This manuscript is a review about the role that NRF2 plays in cancer and its blockage as a possible way for treatment.
The topic is very interesting and useful as it is important to have deep knowledge of the multiple mechanisms of carcinogenesis in order to understand and find therapeutic strategies
However, I consider that the text is too dense. I acknowledge that the topic is complex, but I would suggest to rewrite the manuscript trying to avoid too length paragraphs, or perhaps diving them in more sections.
Moreover, the figure that could be of help it doesn’t. the sequence of the different steps it is not clear. Also, NRF2 in the figure is represented as Nrf2.
I think that the manuscript would benefit if some understanding figure/es was/were incorporated
Author Response
Response to Reviewer 1
Comment: “The text is too dense… rewrite to avoid very long paragraphs or divide into more sections.”
Response: We thank the reviewer for this valuable comment and fully agree. Accordingly, the manuscript was reorganized into more subsections, with long blocks of text divided into shorter, clearer paragraphs to improve readability and narrative flow.
Comment: “The figure could be helpful but is not; the sequence of steps is unclear. NRF2 is inconsistently labeled as Nrf2.”
Response: We thank the reviewer and agree with this observation. We redesigned Figure 1 to clarify the sequence of events in KEAP1–NRF2 regulation and corrected terminology inconsistencies throughout the figure and manuscript.
Comment: “The manuscript would benefit from incorporating understanding figure(s).”
Response: We appreciate this suggestion and agree. The figure was improved to serve as an intuitive visual guide to support key mechanisms discussed in the text.
Reviewer 2 Report
Comments and Suggestions for Authors
The review article is interesting, but some modifications are recommended.
- Title should be modified to be clearer. It can be understood from the title that inhibition of NRF2 is implicated in cancer progression, but the idea of the review is the opposite of that.
- In introduction, clarify the structure homology and domains of NRF2 because some of the drugs or inhibitors block specific domain in the structure.
- In introduction or in relation in cancer, the authors should clarify that NRF2 acting as a "double-edged sword" i.e. clarify that NRF2 can be protective and oncogenic.
- A new heading about crosstalk or interactions with other common signalling pathways involved in cancer development should be added to make it easier understand the next section of therapeutics.
- The mentioned therapeutics should be summarized in tables with their corresponding reference.
- Why CUL3 mutations involved in tumors not mentioned in table 1?
- Add heading about future prospectives.
Author Response
Response to Reviewer 2
Comment: “The title should be modified to be clearer… in the original version it could be interpreted that NRF2 inhibition promotes cancer progression.”
Response: We thank the reviewer for this important clarification and agree with the concern. The title was revised to more accurately reflect the focus of the review.
Comment: “In the introduction, clarify the structure homology and domains of NRF2, since some inhibitors target specific domains.”
Response: We thank the reviewer and fully agree. The introduction now includes the most frequent mutated motifs of the Kelch domain
Comment: “Clarify the double-edged sword role of NRF2.”
Response: We thank the reviewer for highlighting this point and agree. The introduction now clearly explains NRF2’s tumor-suppressive effects in early carcinogenesis and its oncogenic functions when constitutively activated. (Line 36-40)
Comment: “Add a new heading about crosstalk with other signaling pathways involved in cancer development.”
Response: We thank the reviewer for this excellent suggestion and agree. In response, we added a new section describing NRF2 interactions with PI3K/AKT, MAPK, TGF-β, and NF-κB pathways. (Line 147-171)
Comment: “Summarize the therapeutics in tables with references.”
Response: We thank the reviewer and fully agree that a table improves clarity. A new table summarizing all NRF2 modulators—including mechanisms, available doses, and references—was added to the therapeutics section. (Line 409)
Comment: “Why are CUL3 mutations not mentioned in Table 1?”
Response: We thank the reviewer for pointing out this omission and agree. CUL3 alterations were added to Table 1 with new references which are 75-78.
Comment: “Add a heading on future perspectives.”
Response: We thank the reviewer for the excellent suggestion and agree. A new “Future Perspectives” section was added, discussing clinical translation, biomarker development, patient stratification, and the opportunities and challenges of targeting NRF2 in oncology. (line 446)
Round 2
Reviewer 1 Report
Comments and Suggestions for Authors
The authors have improved the manuscript and make it more understandable.
Reviewer 2 Report
Comments and Suggestions for Authors
All comments have been addressed and covered